# Platinum-bearing chromite layers are caused by pressure reduction during magma ascent

Rais Latypov [1], Gelu Costin[2], Sofya Chistyakova[1], Emma J. Hunt [1], Ria Mukherjee[1] & Tony Naldrett[3]

Platinum-bearing chromitites in mafic-ultramafic intrusions such as the Bushveld Complex are key repositories of strategically important metals for human society. Basaltic melts saturated in chromite alone are crucial to their generation, but the origin of such melts is controversial. One concept holds that they are produced by processes operating within the magma chamber, whereas another argues that melts entering the chamber were already saturated in chromite. Here we address the problem by examining the pressure-related changes in the topology of a $Mg_2SiO_4$–$CaAl_2Si_2O_8$–$SiO_2$–$MgCr_2O_4$ quaternary system and by thermodynamic modelling of crystallisation sequences of basaltic melts at 1–10 kbar pressures. We show that basaltic melts located adjacent to a so-called chromite topological trough in deep-seated reservoirs become saturated in chromite alone upon their ascent towards the Earth's surface and subsequent cooling in shallow-level chambers. Large volumes of these chromite-only-saturated melts replenishing these chambers are responsible for monomineralic layers of massive chromitites with associated platinum-group elements.

[1] School of Geosciences, University of the Witwatersrand, Johannesburg, 2050, South Africa. [2] Department of Earth Science, Rice University, Houston, 6100 TX, USA. [3] Department of Earth Sciences, University of Toronto, Toronto, 1066, Canada. Correspondence and requests for materials should be addressed to R.L. (email: rais.latypov@wits.ac.za)

The origin of platinum-bearing chromitites (up to 90 vol.% chromite), which occur as monomineralic layers up to a few metres thick and hundreds of kilometres in length within mafic-ultramafic intrusions (Fig. 1), has been under intense scientific scrutiny since the early twentieth century[1–13]. This persistent interest is driven not only by the challenge of resolving the thermodynamic reasons for their formation[2–5,8], but also by their exceptional economic significance—they host over 90% of Earth's resources of chromium, an important industrial element[1] for improving the physical and chemical properties of steels. Some massive chromitites are also highly enriched in platinum-group elements[1,8], which are industrially used to break down toxic exhaust gases into relatively benign species. Historically, much of this attention has been focused on the Bushveld Complex—the largest mafic magmatic body on Earth—which hosts most of the world's resources of chromite and the platinum-group elements. Despite its enormous size, this complex is still merely the uppermost portion of the much larger transcrustal magmatic system that transferred mantle-derived melts through several staging reservoirs towards the Earth's surface[14]. There is a growing understanding that the solution to the origin of massive chromitites in the Bushveld Complex and other layered intrusions, must be sought in the mechanisms via which magma is transferred through the crust[8–12].

It is widely believed that massive chromitites are best explained by crystallisation from mantle-derived basaltic melts, in which chromite was the only liquidus phase. The major challenge is, however, to explain how exactly the Earth's transcrustal magmatic system generates melts capable of forming chromite deposits. Traditionally, melts directly derived from the mantle or deep-seated magma reservoirs are not thought to be capable of being saturated in chromite alone. This is because olivine is a residual phase in the mantle and its stability field expands with decreasing pressure. Therefore, ascent of mantle-derived magmas towards the Earth's surface will ensure their saturation in olivine (±chromite), rather than chromite alone. For this reason, various mechanisms: including magma mixing; an increase in total pressure or $fO_2$ of the magma; or contamination by crustal rocks[2–7] have been proposed to generate melts saturated in chromite alone within shallow crustal magma chambers. Recently, an intriguing alternative hypothesis has been developed. This attributes generation of the voluminous Bushveld chromitite layers to crystallisation from mantle-derived magmas, which became saturated with chromite alone prior to their emplacement into the chamber[8]. Although this novel idea is strongly supported by textural and field observations from other mafic-ultramafic-layered intrusions and sills[9–12,15,16], the exact reason for saturation of mantle-derived magmas in chromite alone, has not yet been identified.

Here, we present an original solution to this challenging problem using three interrelated approaches—phase equilibria, thermodynamic modelling and experimental data. We show that chromite alone saturation of basaltic magmas is likely due to a previously unrecognised topological feature of liquidus phase equilibria—a so-called chromite topological trough. Basaltic melts located alongside this trough in deep-seated reservoirs may become saturated in chromite alone upon their transcrustal ascent towards the Earth's surface. This occurs due to expansion of the plagioclase stability volume as the lithostatic pressure decreases, which shifts the chromite topological trough, forcing these basaltic melts into the chromite stability volume. These melts would crystallise pure chromite on cooling forming massive chromitites in layered intrusions. This discovery dramatically changes our understanding of the petrogenesis of world-class chromitite and platinum deposits, while introducing a novel approach for interpreting other magmatic deposits.

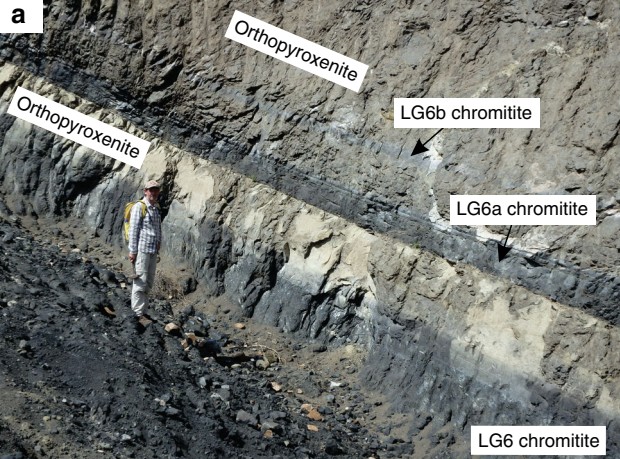

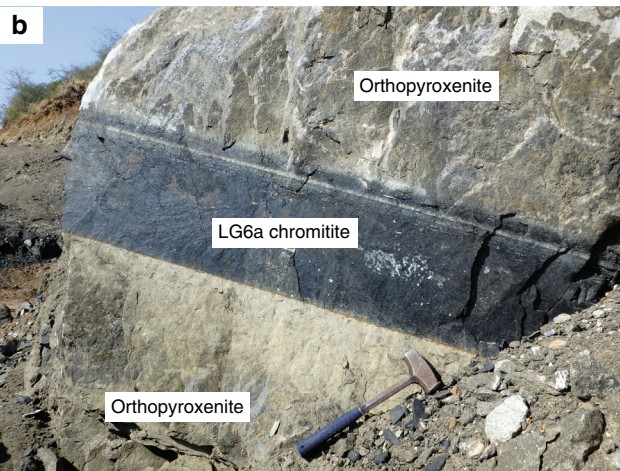

**Fig. 1** Massive chromitite layers of the Bushveld Complex, South Africa. **a** LG6, LG6a and LG6b massive chromitites exposed in an open pit mine working from the Cameron section, Eastern Bushveld Complex. Person for scale is ~1.85 m. **b** A closer view of the LG6a massive chromitite within the same open pit mine working as above (Cameron section, Eastern Bushveld Complex). The hammer for scale has a length of about 50 cm

## Results

**Graphical phase equilibria.** Ascent of basaltic melts from a mantle source region results in lithostatic pressure reduction, which may substantially affect the liquidus phase equilibria. These changes have marked consequences on melt crystallisation sequences, as illustrated here by pressure-induced modification of the quaternary system $Mg_2SiO_4$–$CaAl_2Si_2O_8$–$SiO_2$–$MgCr_2O_4$ (Fig. 2a). In this diagram, we have identified one crucial feature that has so far not been recognised. This is the chromite topological trough that is located between the orthopyroxene/olivine and plagioclase stability volumes (Fig. 2b). The trough results from a substantial decrease in the proportion of cotectically crystallising chromite, as one moves from invariant points in ternary systems (1.4–1.5 wt. and 0.3 wt.%) towards quaternary ones within a tetrahedron (0.12–0.18 wt.%). With decreasing lithostatic pressure, the plagioclase volume expands at the expense of the orthopyroxene and partly olivine volumes[17], thus shifting the chromite topological trough away from the plagioclase apex (Fig. 3a). We consider the effect of this change in the liquidus topology for initially orthopyroxene-saturated melts. The most primitive melts, located far away from the trough, will remain orthopyroxene-saturated at both high and lower pressures (Fig. 3b, c, path from A to B). However, some evolved orthopyroxene-saturated melts, which at high pressures are

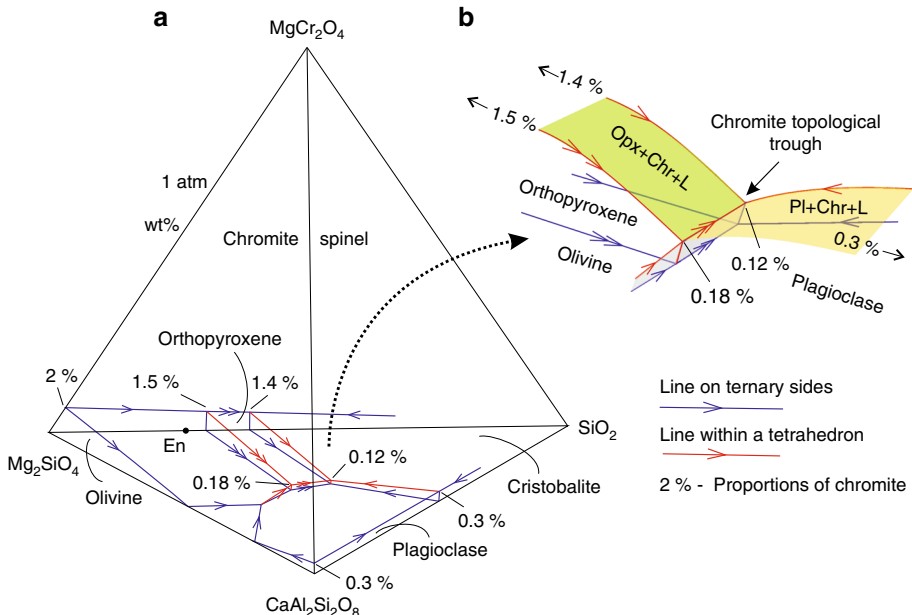

**Fig. 2** The chromite topological trough in basaltic systems. **a** Phase relations in a quaternary system $Mg_2SiO_4$–$CaAl_2Si_2O_8$–$SiO_2$–$MgCr_2O_4$ at 1 atm. The stability volume of chromite, represented by a complete solid solution series between $MgAl_2O_4$-spinel and $MgCr_2O_4$-spinel, occupies almost the entire interior of the diagram. It is continuous between the top apex of the tetrahedron and the small triangular liquidus field of $MgAl_2O_4$-spinel on the basal ternary diagram $Mg_2SiO_4$–$CaAl_2Si_2O_8$–$SiO_2$. **b** A close-up of the diagram showing the chromite topological trough between stability volumes of orthopyroxene/olivine and plagioclase. The trough plays a decisive role in the generation of basaltic melts saturated in chromite alone (Fig. 3). The figure is modified from ref. [13], with the topology being slightly distorted for illustration purposes. Here and in all subsequent figures, cotectic and reaction lines are indicated by one or two arrows, respectively. Here and in all subsequent figures and text: An, anorthite; Fo, forsterite; Opx, orthopyroxene; Ol, olivine; Pl, plagioclase; Cpx, clinopyroxene; Chr, Cr-spinel; Qtz, quartz; Grt, garnet

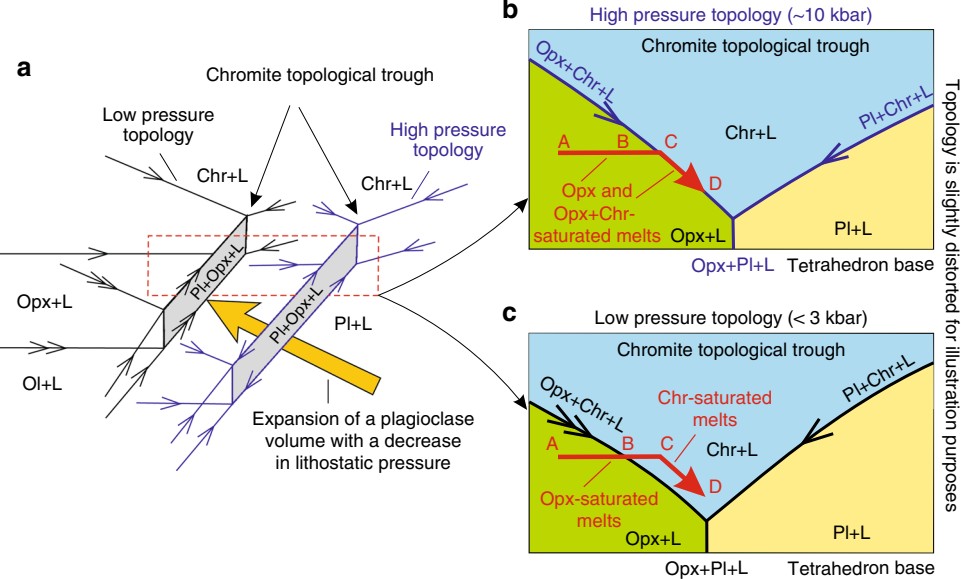

**Fig. 3** A liquidus phase diagram illustrating the mechanism via which melts become saturated in chromite alone by pressure reduction. **a** Expansion of the plagioclase stability volume with decreasing pressure results in a shift of the chromite topological trough away from the plagioclase apex. **b** A high pressure section through the chromite topological trough and related stability volumes showing a range of basaltic melts saturated in orthopyroxene (from A to C) and orthopyroxene + chromite (from C to D). **c** The same section at lower pressures indicates that the most primitive melts (from A to B) will remain orthopyroxene-saturated over the entire pressure range. However, the less primitive orthopyroxene-saturated melts (from B to C) and all orthopyroxene + chromite-saturated melts (from C to D) will move into the chromite topological trough at lower pressure (e.g. ~0–3 kbar). Note that in this process the chromite stability field does not expand but merely changes its position. The location of the section (**b**, **c**) is indicated by a rectangle in **a**, this rectangular section is not a temperature-composition binary diagram

| Table 1 Composition of basaltic liquid used for alphaMELTS modelling at a pressure range from 1 to 10 kbar | | | | | | | | | | | |
|---|---|---|---|---|---|---|---|---|---|---|---|
| Composition | $SiO_2$ | $TiO_2$ | $Al_2O_3$ | $Fe_2O_3$ | $Cr_2O_3$ | FeO | MgO | CaO | $Na_2O$ | $K_2O$ | $H_2O$ | Total |
| Basaltic liquid | 55.05 | 0.11 | 17.41 | 0.90 | 0.10 | 7.91 | 6.69 | 9.83 | 1.44 | 0.11 | 0.45 | 100.00 |

located close to the trough (Fig. 3b, path from B to C) and all orthopyroxene-chromite melts located on the trough (Fig. 3b, path from C to D), will at lower pressures be forced into the trough, becoming saturated in chromite only (Fig. 3c). Similar behaviour is expected for olivine- or olivine-chromite saturated melts. Thus, mantle-derived basaltic melts located alongside the chromite topological trough may become saturated in chromite alone during transcrustal transfer towards the Earth's surface.

**Thermodynamic modelling.** We have tested the above prediction of a graphical approach by thermodynamic modelling, using the alphaMELTS software[18,19] (Methods), with a focus on generation of chromitites from the Critical Zone (CZ) of the Bushveld Complex[8]. These rocks are inferred to have parental melts that are pyroxenitic to noritic in composition[20,21], which are produced by contamination and fractional crystallisation of primary ultramafic magmas in staging chambers[22,23]. Starting from one of these compositions (B2 melt[21], Table 1) we incrementally and iteratively modified its major element contents (Methods) to explore whether isobaric crystallisation of the obtained basaltic melts may identify a pressure interval within which chromite is the sole liquidus phase. The modelled basaltic liquid that provides the best insights into the crystallisation sequence within a pressure interval from 10 to 1 kbar is listed in Table 1. This basaltic liquid (MgO = 6.68 wt.%; $Cr_2O_3$ = 0.10 wt.%) is saturated in both orthopyroxene and chromite along the pressure interval from 10 to 6 kbar (Fig. 4) but has chromite as the first liquidus phase, followed by orthopyroxene at pressure lower than 6 kbar. With decreasing pressure, the temperature interval between the first appearance of liquidus chromite and orthopyroxene increases by up to 40 °C at 1 kbar. Note also that the temperature difference between the appearance of liquidus chromite/orthopyroxene and plagioclase reduces in response to the progressive expansion of the plagioclase stability volume as the pressure decreases (Fig. 4). Thus, the results of this modelling entirely corroborate the key inference of our graphical phase equilibria analysis: that melt compositions located on the chromite topological trough at high pressure become purely saturated in chromite at low pressure (Fig. 3, path C–D).

**Experimental data.** We have further tested the prediction of graphical and thermodynamic approaches by re-examining the available experimental data on chromite stability in basaltic systems. There are not yet fully appreciated data that provide direct support for our theoretical modelling. In particular, data for Cr-solubility in basaltic melts at chromite saturation clearly indicate that the chromite stability volume expands as pressure decreases[24,25]. These data should not be confused with experiments indicating that the spinel stability volume, in contrast, shrinks with decreasing pressure at the expense of the silicate minerals[17]. These experiments are done in simple Cr-free and Fe-free systems and are therefore not directly applicable for the behaviour of chromite stability volume in multicomponent basaltic liquids. The resultant change in the position of the chromite topological trough (Fig. 3c), would contribute to moving some basaltic melts, initially saturated in olivine/orthopyroxene and chromite, directly into the chromite stability volume in response to a decrease in lithostatic pressure. Even more importantly, several experimental studies directly indicate that some natural basaltic magmas, at a

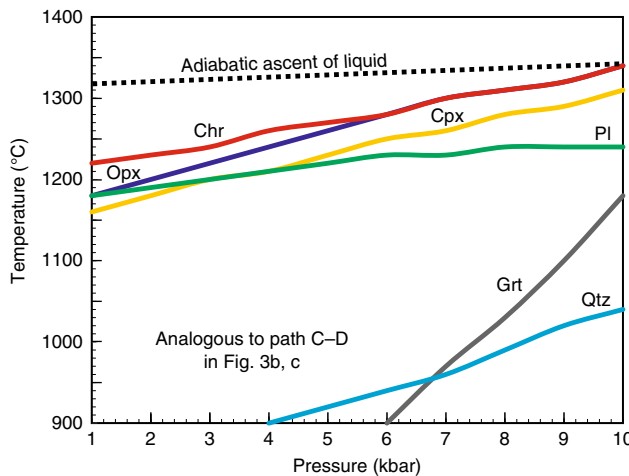

**Fig. 4** Thermodynamic modelling indicating the mechanism via which basaltic melts become saturated in chromite alone by pressure reduction. The basaltic liquid that is saturated in orthopyroxene and chromite at high pressures (along the interval from 10 to 6 kbar) becomes saturated in chromite alone, which is the first liquidus phase, followed by orthopyroxene, at low pressures (P < 6kbar). This case is analogous to path C–D in Fig. 3b, c. Note that adiabatic ascent of the crystal-free melts from a deep-seated storage region located at pressure of ~10 kbar towards the Earth's surface may result at 1–3 kbar in up to 80–95 °C of magma superheating relative to their liquidus temperature. Crystallisation of the basaltic liquid (Table 1) was modelled at FMQ oxygen buffer and a low water content (0.45 wt.%) using alphaMELTS software, version 1.4.1[18, 19]. The results are summarised in the electronic appendix (Supplementary Data 1 and 2)

wide range of water contents and oxygen fugacities, have chromite as the primary liquidus phase at low pressures[26–28]. As an illustration, anhydrous phase relations for a primitive arc basalt (MgO = 15.13 wt%; $Cr_2O_3$ = 0.10 wt%)[28] indicate that at high pressures (6–10 kbar) it is saturated in both olivine and chromite, whereas at low pressures (0–6 kbar) chromite becomes the only liquidus phase (Fig. 5b). This pure chromite saturation apparently occurs because the initial melt composition is, at high pressures, located on the olivine-chromite liquidus surface of the chromite topological trough (Figs. 2 and 3).

## Discussion

The above three lines of evidence—graphical phase equilibria, thermodynamic modelling and experimental data—collectively indicate that mantle-derived basaltic melts lying close to, or directly on, the chromite topological trough, may become saturated in chromite alone in response to a decrease in lithostatic pressure. This finding provides a basis for our novel proposal for the origin of massive chromitites in layered intrusions (Fig. 5). We suggest that such basaltic melts, derived either directly from the mantle or deep-seated crustal reservoirs, become first superheated during their upward ascent. For instance, adiabatic ascent of melts from a deep-seated storage region at ~10 kbar towards a shallow-level intrusion at ~1–3 kbar may result in about 80–95 °C of magma superheating (Fig. 4). The adiabatic path may be favoured by rapid ascent of large volumes of magma from depth

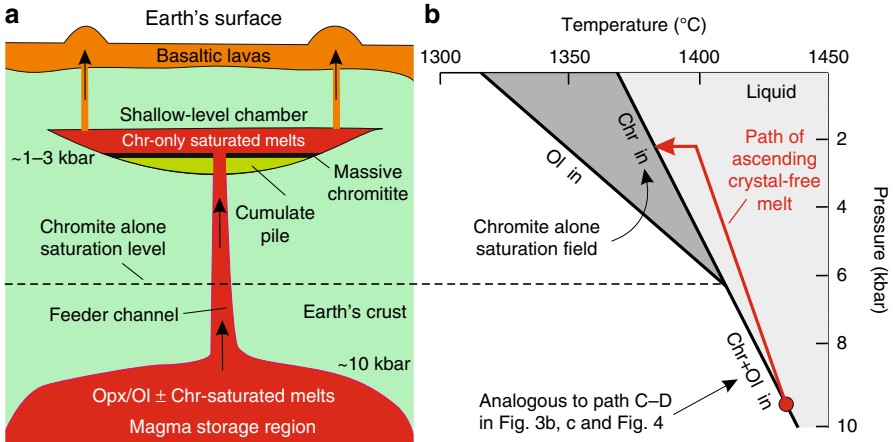

**Fig. 5** A physical model for the generation of basaltic melts saturated in chromite alone by a reduction in lithostatic pressure. **a** Mantle-derived basaltic melts ascending from lower crustal storage regions, or a mantle source, inevitably experience a reduction in lithostatic pressure. This results in shifting of the chromite topological trough and, as a result, basaltic melts located alongside the chromite topological trough, at high pressure regions will become saturated in chromite alone during ascent to shallow-level chambers. Fractional crystallisation of a large volume of these chromite-saturated melts, in an open system where magma can also flow out of the chamber, will produce monomineralic layers of massive chromitites in mafic-ultramafic intrusions. **b** Phase relations for a primitive basalt (MgO = 15.13 wt.%; $Cr_2O_3$ = 0.10 wt.%) in $P$–$T$ space illustrating the model that basaltic melts located alongside the chromite topological trough first become slightly superheated during their ascent and then saturated in chromite alone after stalling and cooling in shallow-level chambers. Therefore, allowing for the development of massive chromitites in shallow-level chambers. This case is analogous to path C–D in Fig. 3b, c and Fig. 4c, in which multiply-saturated liquids become saturated in chromite alone with pressure reduction. The phase diagram is simplified from Fig. 10 in ref. [28] and is only used to graphically illustrate the principle lying at the heart of our model

so that its cooling would be negligible. In reality, some cooling of magma will inevitably occur in response to resorption of any suspended phenocrysts inherited from the deeper magma reservoir, assimilation of crustal wall rocks, and conductive heat loss to cold crustal rocks. The path shown in Fig. 5b takes into account these possible cooling mechanisms during ascent so that upon arrival into a shallow-level chamber the magma will only be superheated by ~10 °C (Fig. 5b). Simple calculations illustrate that magmas with such a degree of superheating are capable of thermochemically eroding a substantial amount of pre-existing footwall cumulates[10–12,29]. This may explain why most, if not all, of the Bushveld's massive chromitites regionally cut down through several metres of the underlying cumulates and overly regionally extensive igneous unconformities[12].

After erosion of footwall rocks with resultant cooling, the shallowly emplaced magma starts to crystallise chromite alone (Fig. 5b) forming a layer of massive chromitite, with associated platinum-group elements, on the chamber floor (Fig. 5a). We suggest that periodic replenishment of shallow-level crustal chambers by such melts is most likely responsible for the development of massive chromitites in mafic-ultramafic intrusions. The production of thick layers of massive chromitite requires a great enough volume of chromite-saturated magma. This is not an issue as most intrusions that host chromitites acted as open systems, with large volumes of magma (equivalent to a layer several km thick) flowing in and out of the system[7,8]. In particular, there is evidence for this open system behaviour in the Bushveld Complex, with vast quantities of inflowing magmas being subsequently expelled from the chamber and now locally preserved as basaltic lavas of the Rooiberg Group[22,30]. An advantage of our proposal is that it does not require the complex internal processes that are commonly invoked for chromite-saturation within upper crustal chambers[2–7]. Scrutiny of the concepts involving these processes during the last few decades has revealed that none can adequately explain characteristic features of massive chromitites[8,20,31–33]. Similarly, our proposal makes unnecessary a currently popular appeal for the formation of massive chromitites from crystal-rich slurries, through

mechanical segregation of chromite from modally prevailing olivine and orthopyroxene[31–33], which is associated with several serious physical and compositional problems[6,12,20].

The major strength of our novel proposal is in its inherent simplicity. All ascending mantle-derived melts on our planet are inevitably subjected to a reduction in lithostatic pressure. Therefore, it is quite natural that some of them, compositionally located alongside the chromite topological trough, may reach chromite only saturation. This scenario appears to be conceptually the simplest and physically most plausible explanation for a generation of chromite-saturated melts parental to massive chromitites in layered intrusions. This new mechanism necessitates profound reassessment of existing petrogenetic models for chromitite deposits. We also infer that this mechanism can be extrapolated to explain the podiform chromitites that are commonly observed in ophiolites as well as non-ophiolitic peridotite complexes[34–36]. In addition, we predict that the origin of other magmatic deposits can ultimately be resolved by examining the consequences of lithostatic pressure reduction on liquidus phase equilibria. This appears to be a quite promising, yet currently poorly explored, line of research. This study thus potentially provides a long-missing piece of the puzzle for our understanding of the Earth's magmatic activity that led to the generation of ore-forming melts. The reduction in lithostatic pressure during the transcrustal transfer of mantle-derived melts may represent an outstanding creator of other types of ore-forming melts (e.g. magnetite-only-saturated) thus developing strategic planetary resources, without which modern human society cannot develop in a sustainable manner[37].

## Methods

**MELTS programme.** Thermodynamic calculations were carried out using alpha-MELTS software, version 1.4.1[18,19]. The implementation in computing software of the thermodynamic properties of Cr-bearing phases, involving partitioning of chromium between melt, ortho- and clinopyroxene, in the presence or absence of spinel and/or garnet, has certain limitations. However, our approach targets close-to-liquidus phase relationships, where clinopyroxene, the main silicate host of chromium, is not stable. Hence, the calculations involving equilibrium between liquid-chromite-orthopyroxene-plagioclase are only associated with small errors,

due to the fractions of percent of $Cr_2O_3$ hosted in orthopyroxene. On the basis of fundamental theories of solid solutions: the effect of $Cr_2O_3$ on the density of orthopyroxene and spinel/chromite and the Cr–Al substitutions in orthopyroxene and chromite, one can predict that chromium partitioning into orthopyroxene in the absence of clinopyroxene and in the presence of spinel/chromite and liquid, will have several effects. Among them are an insignificant lowering of the temperature stability field of Cr-bearing orthopyroxene relative to liquid and chromite at low pressure and a negligible increase of the stability field of Cr-bearing orthopyroxene relative to chromite at high pressures. These effects support the concepts of our approach, where a basaltic magma has liquidus (Cr-bearing) orthopyroxene at high pressures, but (higher aluminium) chromite at lower pressures. These effects therefore indicate that the assumption that the influence of $Cr_2O_3$ in orthopyroxene is negligible is valid.

**Parental melt and its crystallisation sequence**. The MELTS algorithm[38] was used to explore the relative crystallisation sequence of the B2 melt that is commonly considered as parental for the chromitite-bearing CZ of the Bushveld Complex[20,21]. We have tested this melt by running the isobaric calculations at the low pressures proposed for the emplacement of the complex (1–3 kbar)[39]. The results showed that plagioclase is the first mineral to crystallise after chromite, followed by clinopyroxene (pigeonite), and then olivine close to the solidus temperature. No orthopyroxene was produced at low pressures. The outcome is thus in striking contradiction with the phase relationships in the Upper CZ (UCZ) which has orthopyroxene as the first liquidus phase. Therefore, the conventional B2 melt was abandoned in our thermodynamic calculations. Instead, we have modified its major element composition to reproduce the mineral assemblage observed in the UCZ. In particular, the $SiO_2$ was increased to promote the crystallisation of orthopyroxene instead of olivine. The $Al_2O_3$ and CaO were adjusted to reproduce the plagioclase in the sequence of crystallisation and its composition in the Bushveld UCZ. The $Cr_2O_3$ was increased to explore the temperature of chromite crystallisation and reproduce its composition observed in massive chromitites of the Bushveld CZ. FeO content was decreased to allow orthopyroxene crystallisation before plagioclase and clinopyroxene, to assure clinopyroxene crystallisation after plagioclase, and to increase MgO/FeO ratios in orthopyroxene and chromite, to match their compositions with those in the Bushveld CZ. In this process, we have identified several potential basaltic liquids that produce crystallisation sequences and mineral composition that are roughly comparable with those observed in the Bushveld CZ. One of these liquids is presented in Table 1. Calculations for this basaltic composition were run over a pressure range from 10 to 1 kbar at FMQ oxygen buffer and a low water content (0.45 wt.%) (Supplementary Data 1 and 2). The temperatures of the first appearance of minerals for every run were compiled and plotted in a pressure-temperature diagram.

**Data availability**. The authors declare that all relevant data are available within the article and its supplementary information files.

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

## Acknowledgements

We are grateful to B. Robins, B. O'Driscoll, O. Namur, J. Kramers, L. Ashwal, W. Maier, E. Mathez, S. Barnes, A. Morse, J. Day, C. Ballhaus, N. Arndt, I. Veksler, A. Boudreau, P. Horvath, M. Yudovskaya and I. Campbell for fruitful discussions on many aspects of this study as well as for critical comments and useful suggestions on the numerous earlier versions of this manuscript. The research was supported by several research grants to R. L., S.C., E.H. and R.M. from the National Research Foundation of South Africa.

## Author contributions

R.L, S.C. and T.N. conceptualised the original idea. R.L. and S.C. constructed phase diagrams and wrote the original draft paper. G.C. performed thermodynamic calculations. E.H. and R.M. participated in data processing and interpretation as well as in improving clarity of a table and figures. All co-authors discussed the results and problems and contributed to producing a final draft for peer reviews and in revising the manuscript after peer and official reviews.

## Additional information

**Competing interests:** The authors declare no competing financial interests.

