## [Peer Review File · Nature Communications]

Reviewers' comments:

Reviewer #1 (Remarks to the Author):

This is a very interesting manuscript that provides a possible solution to a longstanding geological problem – the origin of chromite deposits in the Bushveld complex. As explained by the authors, these deposits have great economic importance, being the source of a major part of the world's chrome. To account for their existence requires a better understanding of fundamental processes that affect transfer of magma from mantle sources into the continental crust. The authors' solution is convincing and well explained and in my opinion the manuscript should be published in Nature.

I have a number of comments and suggestions:

line 62: it is intriguing that the trough was not recognized by others since the phase diagram was published some 40 years ago. Maybe the authors could comment on this.

line 80: the pyroxenitic or noritic composition of proposed parental Bushveld magmas is the result of large amounts of contamination of primary ultramafic magmas. Some discussion is required of the evolution of these primary mantle melts to the more evolved melts represented in Figure 2. I think the authors' model would work if the contamination took place in the lower crust, as advocated by Maier et al. (2000). Then the pressure drop required to move the melt into the chromite field could have happened during transfer from lower to upper crust. A key question is whether a pressure drop of about 10 kbar, corresponding to a 30 km ascent, is sufficient to drive the composition far enough in the chromite field. Is there any way this could be quantified?

line 109: this choice of melt composition is not very appropriate because 1) arc basalts are not anhydrous and 2) Bushveld did not form in a subduction setting. Could a composition corresponding to crust-contaminated ultramafic magma like the B1 sills be used instead?

line 125: but is this the case for the Bushveld? Is there any evidence that a large volume of magma flowed out of the system? If the incoming magma contained only chromite crystals, could this explain the mass-balance" problem; i.e. the amount of chrome must have come from a volume of magma bigger than that currently present in the intrusion?

Reviewer #2 (Remarks to the Author):

The authors present a model to explain the origin of massive chromitite layers of the Bushveld Complex. The origin of these layers is a long-standing problem. How can a trace mineral like chromite be concentrated to meter-thick, near-monomineralic layers when the melts are co-saturated with olivine or orthopyroxene? It is proposed here based on thermodynamic modelling using the algorithm MELTS that basaltic magmas pre-fractionated at depth may lose their liquidus silicates (here $\text{opx} \pm \text{ol}$) when they decompress. Hence, upon arrival in the Bushveld chamber, the magmas may be saturated with chromite only.

Principally, I find the model presented interesting and worth being considered for publication in Nature Comm. If the model works, it will add another mode of origin for these enigmatic stratigraphic units in layered intrusions in which chromite is the only liquidus phase. However, before I can recommend acceptance, the authors should address the points listed below.

Chromite topological trough. It may not be so clear to some readers why that deep-seated magma chamber is needed. I assume the melts derived from the mantle will have to be pre-fractionated at depth to become opx -chromite-saturated, i.e. path A-C in Fig. 2b? Can chromite- only saturated melts also be generated by decompressing melts straight from the mantle, as soon as they enter the continental crust? Note that the melts proposed in Table 1 are very magnesian. If these

compositions are realistic (see below) they do not look like pre-fractionated based on their MgO contents. The first major oxide that will be depleted by crystal fractionation is MgO. Or are the compositions listed in Table 1 those melts that are assumed to have been added to the deep-seated magma reservoir prior to fractionation?

Melt compositions used for the model. I wonder if the compositions listed in Table 1 are realistic melts. There are no mantle melts or derivative melts from basalts reported with > 13 wt.% MgO and only 4.5 wt.% FeO. I assume these are chilled margin compositions communicated by Wolf Maier in *Econ Geol*? Presently I have no access to that paper, so I cannot check. I have calculated the liquidus olivine compositions that would crystallize from the melts 1 and 2 using a KD (Fe-Mg, ol-melt) of 0.3 (the accepted value): Fo95. No primitive melt would have Fo95 on the liquidus unless it is extremely oxidized (~ MH). The ferric/ferrous ratios of the compositions quoted in Table 1 are not as oxidized, they are around a relative fO₂ of FMQ. Even if opx is the first liquidus silicate (likely given the high SiO₂ contents), opx would still be as magnesian as olivine because the KD (ol-opx) at magmatic temperature is near unity. My recommendation is to use a more realistic melt composition, a melt that has been studied experimentally, and include some water content. The Bushveld melts may have been H₂O-enriched. Water may stabilize chromite relative to silicates.

I assume the path A-C in Fig. 2a is somewhat schematic. Nonetheless, the authors should try to semi-quantify how much MgO a melt would lose by fractionating opx±ol from points A to C, then consider how realistic their compositions in Table 1 are.

Ascent path. The authors need to specify how the superliquidus ascent path in Fig. 4b was constructed. Is that path an adiabat? The authors should calculate the adiabat for their melt compositions because it defines the minimum temperature drop during ascent. There are three factors that may prevent ascending melts from following a near-adiabatic ascent path and ascend at superliquidus temperatures; (1) resorption of phenocrysts in suspension inherited from the deeper magma reservoir which consumes latent heat of fusion, (2) wall rock assimilation, and (3) heat loss by conduction to the crust. Note that the crust through which these melts must rise is relatively cool, much cooler than the liquidus temperature of the mafic melts (1290°C for melt 2 at 1 atm). These melts will have to rise for around 20 km (Fig. 4) through the crust before they reach the Bushveld chamber proper. That may prove a real challenge for a superliquidus melt to remain at superliquidus temperature. For the model proposed, do the melts have to ascend at superliquidus temperature? If that is not the case, the model should perhaps be amended accordingly to make it more credible.

Discharge mechanisms. Will melts residing in the deeper magma chamber be able to rise up to a shallower magma reservoir so easily, given that the density of the lower crust is less than the density of the melts listed in Table 1? The melt densities may be calculated using algorithms published by Bottinga and coworkers. Are replenishment episodes at depth the driving force?

Chromite-only field. An essential element of the model is the proposition that chromite will be the only liquidus phase at $P < 0.6$ GPa (Fig. 4). A similar diagram was published by Stamper et al. (2014). The resemblance of Fig. 4b of this manuscript with Fig. 10 published by Stamper et al. is striking even though the melt studied by Stamper et al. is quite different to the compositions listed in Table 1. Or was that Fig. 4 adapted from Stamper et al. only to illustrate graphically the principle? If that is so, it would be entirely reasonable, but it should be stated in the figure caption. In Fig. 3b the interval between the chromite and the opx liquidi is very narrow, just a few degrees at 0.3 GPa - is that interval sufficient to accumulate meter-thick chromitites laterally over hundreds of km? In Fig. 4B that interval is ~ 60°C wide, so there is some discrepancy between 3b and 4b that may need be addressed. Note that the melts studied by Stamper et al. were extremely oxidized, close to MH. Will the diagram be applicable to the Bushveld if the relative fO₂ is around FMQ? The FeO/Fe₂O₃ ratios of melt 2 listed in Table 1 are around FMQ. At highly oxidized conditions spinel will be stabilized because high relative fO₂ stabilizes the FeO_{1.5} component which is highly compatible with chromite.

For me as an experimentalist, the most straightforward way would be to define a realistic bulk composition, carry out classic liquidus experiments at FMQ with some H₂O added, and map out the possible extent of a chromite-only field at 1, 0.7, 0.5, and 0.2 GPa. Perhaps use melt 2, but do add some FeO, so much that the liquidus silicate has a reasonable Mg# below Fo90 or En90.

If the manuscript is accepted, I recommend enclosing a figure showing typical outcrops of chromitite layers. That would render the contribution visibly more attractive. I recall from my own field work that the UG chromitite layers may carry silicate phases that look distinctly skeletal, perhaps indicating rather rapid cooling after the melts were emplaced within the Bushveld lithostratigraphy. Skeletal crystals would be supportive for the model presented because they may indicate rapid cooling.

Reply to reviewer's comments:

Reviewer #1 (Remarks to the Author):

This is a very interesting manuscript that provides a possible solution to a longstanding geological problem – the origin of chromite deposits in the Bushveld complex. As explained by the authors, these deposits have great economic importance, being the source of a major part of the world's chrome. To account for their existence requires a better understanding of fundamental processes that affect transfer of magma from mantle sources into the continental crust. The authors' solution is convincing and well explained and in my opinion the manuscript should be published in Nature.

I have a number of comments and suggestions:

line 62: it is intriguing that the trough was not recognized by others since the phase diagram was published some 40 years ago. Maybe the authors could comment on this. **Yes, this is curious. We have now mentioned this in the text.**

line 80: the pyroxenitic or noritic composition of proposed parental Bushveld magmas is the result of large amounts of contamination of primary ultramafic magmas. Some discussion is required of the evolution of these primary mantle melts to the more evolved melts represented in Figure 2. I think the authors' model would work if the contamination took place in the lower crust, as advocated by Maier et al. (2000). **Yes, this makes sense. We have now mentioned this possibility in the paper.**

Then the pressure drop required to move the melt into the chromite field could have happened during transfer from lower to upper crust. A key question is whether a pressure drop of about 10 kbar, corresponding to a 30 km ascent, is sufficient to drive the composition far enough in the chromite field. Is there any way this could be quantified? **Basically, our Fig. 4 (MELTs modelling) has quantified this process and shows that it is sufficient to drive the composition into the chromite stability field (after some cooling in the chamber).**

line 109: this choice of melt composition is not very appropriate because 1) arc basalts are not anhydrous and 2) Bushveld did not form in a subduction setting. Could a composition corresponding to crust-contaminated ultramafic magma like the B1 sills be used instead? This is true, but please, note that Fig. 4b (reproduced from Fig. 10 in ref. 25, Stamper et al., 2014) is only used to illustrate graphically the principle lying at the heart of our model. Of course, it would be much better to use B1 sills magmas but there are currently no experimental P-T data for these compositions. For this reason, for the Bushveld's magmas we have used MELTs modelling (Fig. 3) with low water contents, thus we are not assuming an anhydrous magma.

line 125: but is this the case for the Bushveld? Is there any evidence that a large volume of magma flowed out of the system? Yes, there is. The Rooiberg Group basalts are thought to be (geochemically) those that escaped from the Bushveld chamber (e.g. Maier et al., 2000). The volume of these basalts are not that great but most of them were likely eroded away by deep erosion of the Kaapval craton during the Cretaceous (Partridge, 1998). If the incoming magma contained only chromite crystals, could this explain the mass-balance problem; i.e. the amount of chrome must have come from a volume of magma bigger than that currently present in the intrusion? Yes, sure, a large volume of magma must have been lost from the chamber to satisfy a Cr mass balance issue. We discuss this issue in the paper.

Reviewer #2 (Remarks to the Author):

The authors present a model to explain the origin of massive chromitite layers of the Bushveld Complex. The origin of these layers is a long-standing problem. How can a trace mineral like chromite be concentrated to meter-thick, near-monomineralic layers when the melts are co-saturated with olivine or orthopyroxene? It is proposed here based on thermodynamic modelling using the algorithm MELTS (not entirely true, the idea initially came from phase diagram studies and is only confirmed by MELTS and, quite unexpectedly, by experimental data) that basaltic magmas pre-fractionated at depth may lose their liquidus silicates (here $\text{opx} \pm \text{ol}$) when they decompress. Hence, upon arrival in the Bushveld chamber, the magmas may be saturated with chromite only.

Principally, I find the model presented interesting and worth being considered for publication in Nature Comm. If the model works, it will add another mode of origin for these enigmatic stratigraphic units in layered intrusions in which chromite is the only liquidus phase. However, before I can recommend acceptance, the authors should address the points listed below.

Chromite topological trough. It may not be so clear to some readers why that deep-seated magma chamber is needed. I assume the melts derived from the mantle will have to be pre-fractionated at depth to become opx-chromite-saturated, i.e. path A-C in Fig. 2b? Can chromite- only saturated melts also be generated by decompressing melts straight from the mantle, as soon as they enter the continental crust? **Yes, this is right, principally it does not matter whether the liquids come straight from the mantle (being olivine- or olivine-chromite saturated) or from the staging chamber (being opx- or opx-chromite saturated due to fractionation/contamination) – the result is expected to be the same. We mentioned this in the original paper but have now made it clearer in the revised version.**

Note that the melts proposed in Table 1 are very magnesian **Agree, we have now used melts with much less MgO (6.68 wt% versus 15-10 wt% as in previous compositions) close to B2 magma. The results in terms of crystallization sequences are still the same, however.** If these compositions are realistic (see below) they do not look like pre-fractionated based on their MgO contents. The first major oxide that will be depleted by crystal fractionation is MgO. Or are the compositions listed in Table 1 those melts that are assumed to have been added to the deep-seated magma reservoir prior to fractionation? **Due to the new modelling the table has been updated and should now be clearer to readers. The full results are now located in the supplementary files.**

Melt compositions used for the model. I wonder if the compositions listed in Table 1 are realistic melts. There are no mantle melts or derivative melts from basalts reported with > 13 wt.% MgO and only 4.5 wt.% FeO. **Ok, agree, we have now used one composition that is closer to B2 magma (but not exactly the same – our composition has higher SiO₂ and lower FeO) with 6.68 wt% MgO and 8.80 wt% FeO. This composition appears to be quite realistic. Note that the results in terms of crystallization sequences are**

still the same. I assume these are chilled margin compositions communicated by Wolf Maier in Econ Geol? Presently I have no access to that paper, so I cannot check. I have calculated the liquidus olivine compositions that would crystallize from the melts 1 and 2 using a KD (Fe-Mg, ol-melt) of 0.3 (the accepted value): Fo95. No primitive melt would have Fo95 on the liquidus unless it is extremely oxidized (~ MH). The ferric/ferrous ratios of the compositions quoted in Table 1 are not as oxidized, they are around a relative fO₂ of FMQ. Even if opx is the first liquidus silicate (likely given the high SiO₂ contents), opx would still be as magnesian as olivine because the KD (ol-opx) at magmatic temperature is near unity. Our new basaltic liquid is compatible both with the relative crystallization sequence and mineral composition (e.g. An-content of plagioclase is 82-75%; Mg-number of orthopyroxene=70-72%; Cr₂O₃ of chromite=32-40 wt. %) as observed in the Bushveld CZ. My recommendation is to use a more realistic melt composition, a melt that has been studied experimentally, and include some water content. We failed to find a proper liquid that would have been experimentally studied. However, the new melt does correlate well to the observed compositions in the CZ and has a low water content. The Bushveld melts may have been H₂O-enriched. Water may stabilize chromite relative to silicates.

I assume the path A-C in Fig. 2a is somewhat schematic. Nonetheless, the authors should try to semi-quantify how much MgO a melt would lose by fractionating opx±ol from points A to C, then consider how realistic their compositions in Table 1 are. Yes, this is just schematic and illustrate the principle. I do not therefore feel any additional calculations are necessary here.

Ascent path. The authors need to specify how the superliquidus ascent path in Fig. 4b was constructed. Is that path an adiabat? No, this is not an adiabat. We have calculated adiabatic paths for our compositions and they show that the melts rising from 10 kbar to 1 kbar depth will be superheated by ~ 80-95°C. We have now added this path directly to Fig. 4 with MELTs modelling. Of course, this is an absolute maximum. In reality, the three factors mentioned by the reviewer (below) may result in substantial cooling of the rising magma. Therefore, in Fig. 5b we have indicated a path that shows only about 10°C of superheating which seems quite reasonable. Formation of massive chromitites requires a large volume of magma to be

involved, which would be difficult to cool during its ascent through the crust. Therefore, some superheating of magma will still be preserved after ascent.

The authors should calculate the adiabat for their melt compositions because it defines the minimum temperature drop during ascent. There are three factors that may prevent ascending melts from following a near-adiabatic ascent path and ascend at superliquidus temperatures; (1) resorption of phenocrysts in suspension inherited from the deeper magma reservoir which consumes latent heat of fusion, (2) wall rock assimilation, and (3) heat loss by conduction to the crust. **Yes, this is true and these factors are now mentioned.** Note that the crust through which these melts must rise is relatively cool, much cooler than the liquidus temperature of the mafic melts (1290°C for melt 2 at 1 atm). These melts will have to rise for around 20 km (Fig. 4) through the crust before they reach the Bushveld chamber proper. That may prove a real challenge for a superliquidus melt to remain at superliquidus temperature. For the model proposed, do the melts have to ascend at superliquidus temperature? **The magma superheating is not essential for the model discussed in this paper. But it is important for understanding the origin of Bushveld's massive chromitites in general. The reason for this is that field observations show that footwall silicate rocks were substantially eroded (as indicated by potholes and antipotholes) prior to deposition of chromitites. The best and simplest explanation for this is thermochemical erosion of footwall rocks by superheated magmas as we have discussed in details in our recent paper in JP (Latypov et al., 2017). We have now added a few sentences to address this issue in the paper.** If that is not the case, the model should perhaps be amended accordingly to make it more credible.

Discharge mechanisms. Will melts residing in the deeper magma chamber be able to rise up to a shallower magma reservoir so easily, given that the density of the lower crust is less than the density of the melts listed in Table 1? The melt densities may be calculated using algorithms published by Bottinga and coworkers. Are replenishment episodes at depth the driving force? **I do not feel this issue is problematic. There is plenty of field-based evidence for replenishment of the Bushveld chamber by new pulses of basaltic melts. So it must have occurred. Chromite-only-saturated basaltic magma is not that different from any normal basaltic magmas. It is only slightly oversaturated in chromite.**

Yes, replenishment episodes at depth may be the driving force for magma ascent towards the Earth's surface and emplacement into shallow-level chambers.

Chromite-only field. An essential element of the model is the proposition that chromite will be the only liquidus phase at $P < 0.6$ GPa (Fig. 4). A similar diagram was published by Stamper et al. (2014). The resemblance of Fig. 4b of this manuscript with Fig. 10 published by Stamper et al. is striking even though the melt studied by Stamper et al. is quite different to the compositions listed in Table 1. Or was that Fig. 4 adapted from Stamper et al. only to illustrate graphically the principle? If that is so, it would be entirely reasonable, but it should be stated in the figure caption. **Yes, we have just adapted Fig. 10 from Stamper et al. to illustrate graphically the principle. We do not have experimental P-T data to plot here, for the compositions used for MELTs modelling. We have now indicated this clearly in the paper.** In Fig. 3b the interval between the chromite and the opx liquidi is very narrow, just a few degrees at 0.3 GPa - is that interval sufficient to accumulate meter-thick chromitites laterally over hundreds of km? **Yes, we think so. Please, note that the interval between chromite and chromite+opx crystallization is not that important for the formation of massive chromitites (although of course, the larger the better). What is most important is the volume of chromite-only-saturated magma that is involved, as is clearly stated in the paper. A large amount of chromite-only-saturated magma is required in any model, including ours.**

In Fig. 4B that interval is $\sim 60^\circ\text{C}$ wide, so there is some discrepancy between 3b and 4b that may need be addressed. Note that the melts studied by Stamper et al. were extremely oxidized, close to MH. Will the diagram be applicable to the Bushveld if the relative $f\text{O}_2$ is around FMQ? The $\text{FeO}/\text{Fe}_2\text{O}_3$ ratios of melt 2 listed in Table 1 are around FMQ. At highly oxidized conditions spinel will be stabilized because high relative $f\text{O}_2$ stabilizes the $\text{FeO}_{1.5}$ component which is highly compatible with chromite. **Yes, this is true that the melts studied by Stamper et al. were oxidized and for this reason the interval between chromite and chromite+opx crystallization is so large ($\sim 60^\circ\text{C}$). But once again, we use this P-T diagram for illustration purposes only. For FMQ conditions used in our MELT modelling, this interval is smaller ($\sim 20\text{-}40^\circ\text{C}$) but still more than enough to**

produce massive chromitite given the large volume of chromite-only-saturated magma involved.

For me as an experimentalist, the most straightforward way would be to define a realistic bulk composition, carry out classic liquidus experiments at FMQ with some H₂O added, and map out the possible extent of a chromite-only field at 1, 0.7, 0.5, and 0.2 GPa. Perhaps use melt 2, but do add some FeO, so much that the liquidus silicate has a reasonable Mg# below Fo90 or En90. Agree, it would be nice to do such kind of experiments specifically for the inferred Bushveld's parental magmas. I have a plan to do such experiments in the future (in collaboration with my colleagues).

If the manuscript is accepted, I recommend enclosing a figure showing typical outcrops of chromitite layers. That would render the contribution visibly more attractive. Very good idea, thanks, we have added a figure showing LG6/LG6a chromitites from recent open pits at the Eastern Bushveld. I recall from my own field work that the UG chromitite layers may carry silicate phases that look distinctly skeletal (I have never seen this, interesting...), perhaps indicating rather rapid cooling after the melts were emplaced within the Bushveld lithostratigraphy. Skeletal crystals would be supportive for the model presented because they may indicate rapid cooling.

Summary of major changes in figures:

Fig. 1. A new figure showing massive chromitites of the Bushveld Complex

Fig. 4. Adiabatic ascent path is calculated and added to the figure to illustrate the maximum possible superheating of the melt involved due to pressure reduction.

Fig. 5. We have now shown the cumulate pile and a layer of massive chromitite forming on the temporary chamber floor of the shallow-level intrusion. Also we have shown the magma that flows out of the shallow-level chamber to erupt on the Earth's surface as basaltic lavas.

Best regards,

Rais Latypov

17 October 2017

Principally, I have no objections that this manuscript be published in Nature Comm. The model is interesting. The authors are correct in pointing out the potential significance of decompression on phases relations of basaltic melts, and its potential significance on olivine-chromite stability relations. The devil may lie in the detail though. I suggest the authors consider the points listed below. These points are based largely on the additions to the revised version.

61 *This discovery dramatically changes our understanding of the petrogenesis of world-class chromitite and platinum deposits, while introducing a novel approach for interpreting other magmatic deposits* - that statement may be taken by some readers as an exaggeration. We can shift melts, originally along the olivine-chromite liquidus, by contaminating them with plagioclase cumulates. The process can be visualized easily by using the Fo-An-Di pseudoternary. Melts that assimilate anorthositic footwall will experience exactly that. It might be a good idea to add a sentence where you discuss ways to shift melts into the chromite-only volume.

100 *making the liquid increasingly more saturated in chromite* - better rephrase. A liquid is either saturated or undersaturated. Do you mean to say supersaturated?

110 *Surprisingly, there are unrecognized data* - better rephrase. Surprisingly sounds a little emotional for a scientific paper, unrecognized may also be inappropriate (how do you know).

117 *As an illustration, anhydrous phase relations for a primitive arc basalt (MgO=15.13 wt%; Cr₂O₃=0.10 wt%) indicate that at high pressures (6-10 kbar) it is saturated in both olivine and chromite, whereas at low pressures (0-6 kbar) chromite becomes the only liquidus phase (Fig. 5b)* - This is a critical statement. I suggest you may consult Fig. 18.11 in Morse, Basalts and Phase diagrams (data from Presnall et al. 1978). My impression is that in the simple system, falling pressure will cause the spinel volume to shrink. Note though that the Presnall experiments were carried out without Cr₂O₃, that Cr₂O₃ will affect phase relations greatly. With Cr₂O₃ in a melt, the spinel field will expand, notably at low pressure (for a given melt composition, high pressure spinels are more aluminous than low pressure spinels because increasing pressure causes octahedral sites in silicate melts to increase). Decide yourself how relevant that comment may be, but I do suggest you pay attention to it to avoid criticism from the experimental petrology scene.

128 *This fundamental finding provides a basis for our novel proposal* - may sound a little self-praising

132 *at ~10 kbar towards a shallow-level intrusion at ~1-3 kbar may result in about 80-95°C of magma superheating* - better quote a reference. How did you calculate that ΔT ? What is the adiabat of a basaltic melt?

138 superheated by ~100°C - is that an estimate? It must be because we cannot quantify the heat loss caused by wall rock cooling and assimilation.

144 *After erosion of footwall rocks with resultant cooling, the shallowly emplaced magma starts to crystallize chromite alone (Fig. 5b) forming a layer of massive chromitite* - yes, but do consider that many chromitites of the Bushveld rest on partially resorbed plagioclase bearing cumulates (e.g. Steelpoort), that the assimilation of plagioclase or mixing with plagioclase-saturated resident melt will drive the new melt almost automatically into the chromite-only field (cf. your own Fig. 3b), regardless of your decompression model.

Reply to comments by Reviewer 2

Principally, I have no objections that this manuscript be published in Nature Comm. The model is interesting. The authors are correct in pointing out the potential significance of decompression on phases relations of basaltic melts, and its potential significance on olivine-chromite stability relations. The devil may lie in the detail though. I suggest the authors consider the points listed below. These points are based largely on the additions to the revised version.

61 *This discovery dramatically changes our understanding of the petrogenesis of world-class chromitite and platinum deposits, while introducing a novel approach for interpreting other magmatic deposits* - that statement may be taken by some readers as an exaggeration. We can shift melts, originally along the olivine-chromite liquidus, by contaminating them with plagioclase cumulates. The process can be visualized easily by using the Fo-An-Di pseudoternary. Melts that assimilate anorthositic footwall will experience exactly that. It might be a good idea to add a sentence where you discuss ways to shift melts into the chromite-only volume.

Yes, of course, I am quite aware of this possibility and have even used it myself as an additional explanation for origin of chromite-saturated melts in my previous paper on chromitites of the Merensky Reef (Latypov et al., 2015, *Journal of Petrology*, 56, 2341–2372). I have changed my mind since that time, however, because there are several problems with this popular idea.

First, an idea about contamination of magma by plagioclase-rich cumulates may be potentially applied only to UG1 chromitite which does have anorthosites at its base. However, almost all other chromitites of the Bushveld Complex (LG group, most MG group chromitites and UG2) are sitting among orthopyroxenites and cannot therefore be contaminated by plagioclase-rich cumulates.

Second, Carr et al. (1994, 1999) presented Sr isotopic data that provide no support for in situ contamination of melts parental to the Merensky Reef by footwall anorthosites/norites. Thus this idea, irrespective of how attractive it is, does not work isotopically. I discussed this issue in detail in Latypov et al. (2015).

Third, Mondal and Mathez (2007) presented data for the UG2 sequence that question all models involving mixing of the resident melt with any liquids produced by *in situ* by melting of footwall rocks or introduced into the chamber from outside. They show that any mixing model implies a reversal in composition of minerals above massive chromitite layers. But this was never observed: minerals above and below the chromitite layer have a remarkably similar composition. Again, I am discussing this issue in detail in my paper in press (Latypov et al., 2017, *Journal of Petrology*).

Of course, if you would insist I can add these points to the paper but this will be a repetition of what I have already written in Latypov et al (2015, 2017a, b).

100 *making the liquid increasingly more saturated in chromite* - better rephrase. A liquid is either saturated or undersaturated. Do you mean to say supersaturated? **Good point, agree, this was a bad phrase. I have deleted it.**

110 *Surprisingly, there are unrecognized data* - better rephrase. Surprisingly sounds a little emotional for a scientific paper, unrecognized may also be inappropriate (how do you know).

Ok, I have changed it to the following:
'There are not yet fully appreciated data...'

117 *As an illustration, anhydrous phase relations for a primitive arc basalt (MgO=15.13 wt%; Cr₂O₃=0.10 wt%) indicate that at high pressures (6-10 kbar) it is saturated in both olivine and chromite, whereas at low pressures (0-6 kbar) chromite becomes the only liquidus phase (Fig. 5b)* - This is a critical statement. I suggest you may consult Fig. 18.11 in Morse, *Basalts and Phase diagrams* (data from Presnall et al. 1978). My impression is that in the simple system, falling pressure will cause the spinel volume to shrink. Note though that the Presnall experiments were carried out without Cr₂O₃, that Cr₂O₃ will affect phase relations greatly. With Cr₂O₃ in a melt, the

spinel field will expand, notably at low pressure (for a given melt composition, high pressure spinels are more aluminous than low pressure spinels because increasing pressure causes octahedral sites in silicate melts to increase). Decide yourself how relevant that comment may be, but I do suggest you pay attn to it to avoid criticism from the experimental petrology scene.

Yes, this is true. A field of spinel tends to shrink with falling pressure and this is opposite to the behaviour of chromite stability field. One should not mix these two different sets of experiments because spinel and chromite are two different mineral phases. Experiments on spinel are done in Cr- and Fe-free compositions and have little to do with experiments involving real chromite in basaltic systems. I agree that this potential point for confusion should be mentioned in the paper to avoid criticism from experimental petrologists and petrologists who base their views on the spinel's experiments. This is how I suggest to do this in the text.

We have further tested the prediction of graphical and thermodynamic approaches by re-examining the available experimental data on chromite stability in basaltic systems. There are not yet fully appreciated data that provide direct support for our theoretical modelling. In particular, Cr-solubility data in basaltic melts at chromite saturation, clearly indicate that the chromite stability volume expands as pressure decreases²⁴⁻²⁵. These data should not be confused with experiments indicating that the spinel stability volume, in contrast, shrinks with decreasing pressure at the expense of the silicate minerals¹⁷. These experiments are done in simple Cr- and Fe-free systems and are therefore not directly applicable for the behaviour of chromite stability volume in multicomponent basaltic liquids.

128 *This fundamental finding provides a basis for our novel proposal* - may sound a little self-praising

Ok, a word 'fundamental' can be deleted if you wish:

'This finding provides a basis for our novel proposal...' The proposal is, however, really 'novel'.

132 *at ~10 kbar towards a shallow-level intrusion at ~1-3 kbar may result in about 80-95oC of magma superheating* - better quote a reference. How did you calculate that ΔT ? What is the adiabat of a basaltic melt?

MELTs has an option to calculate adiabatic path of ascending melts. We used this option and results are plotted as a dotted line in Fig. 4. Thus no need for a reference here, it directly comes from Fig. 4. Note that 80 degrees of superheating is for 3 kbar and 95 degrees is for 1 kbar.

138 superheated by ~10oC - is that an estimate? It must be because we cannot quantify the heat loss caused by wall rock cooling and assimilation.

Yes, of course, this is just an estimate that cannot be rigorously constrained. You are right we cannot quantify the heat loss. Some superheating is, however, necessary to explain erosion of footwall rocks as we described in our several publications (Latypov et al., 2015, 2017a, b) and mentioned in this paper.

144 *After erosion of footwall rocks with resultant cooling, the shallowly emplaced magma starts to crystallise chromite alone (Fig. 5b) forming a layer of massive chromitite* - yes, but do consider that many chromitites of the Bushveld rest on partially resorbed plag bearing cumulates (e.g. Steelpoort), that the assimilation of plag or mixing with plag-saturated resident melt will drive the new melt almost automatically into the chromite-only field (cf. your own Fig. 3b), regardless of your decompression model.

Yes, true, theoretically adding of plagioclase-rich melt will help to promote chromite saturation, but, as indicated above, there are serious problems with this idea which we cannot be ignored. The strongest of them is the mineral chemical data presented by Mondal and Mathez (2009). No one has yet shown that they are wrong. Therefore, I feel that we cannot simply promote further magma mixing without addressing their arguments against this idea, not matter how popular it is at the moment. Basically, for this reason I believe that it is much better to use only the pressure-related changes in topology of the phase diagram to make magma chromite-saturated rather than to change a composition of magma itself by in situ mixing/contamination in the chamber.